# Mezigdomide—A Novel Cereblon E3 Ligase Modulator under Investigation in Relapsed/Refractory Multiple Myeloma

**DOI:** 10.3390/cancers16061166

**Published:** 2024-03-15

**Authors:** Monique A. Hartley-Brown, Clifton C. Mo, Omar Nadeem, Shonali Midha, Jacob P. Laubach, Paul G. Richardson

**Affiliations:** Department of Medical Oncology, Dana-Farber Cancer Institute, Jerome Lipper Center for Multiple Myeloma Research, Harvard Medical School, 450 Brookline Avenue, Dana 1B02, Boston, MA 02115, USA; moniquea_hartley-brown@dfci.harvard.edu (M.A.H.-B.); clifton_mo@dfci.harvard.edu (C.C.M.); omar_nadeem@dfci.harvard.edu (O.N.); shonali_midha@dfci.harvard.edu (S.M.); jacobp_laubach@dfci.harvard.edu (J.P.L.)

**Keywords:** CELMoDs, cereblon, mezigdomide, multiple myeloma, relapsed, refractory

## Abstract

**Simple Summary:**

Patients diagnosed with multiple myeloma today can expect to live for substantially longer than in the past thanks to the range of highly active treatment options now available. Over the course of their disease, patients may receive several different treatment combinations to keep their disease under control. Thus, there is an ongoing need for additional new drugs and regimens to be developed, and, in particular, ones that are convenient, accessible, and active against disease that has returned following multiple different therapies. Among today’s standards of care are the immunomodulatory drugs lenalidomide and pomalidomide, which are commonly used in quadruplet and triplet regimens in newly diagnosed patients and those with relapsed disease. However, resistance to these drugs can arise over the course of treatment; therefore, novel, more potent agents are being developed to restore and increase activity against relapsed multiple myeloma, one of which is the investigational agent mezigdomide.

**Abstract:**

Mezigomide is an oral cereblon E3 ligase modulator (CELMoD) that is under clinical investigation in patients with relapsed/refractory (RR) multiple myeloma (MM). Like other CELMoD compounds, mezigdomide acts by altering the conformation of cereblon within the cullin 4A ring ligase–cereblon (CRL4CRBN) E3 ubiquitin ligase complex, thereby recruiting novel protein substrates for selective proteasomal degradation. These include two critical lymphoid transcription factors, Ikaros family zinc finger proteins 1 and 3 (IKZF1 and IKZF3), also known as Ikaros and Aiolos, which have important roles in the development and differentiation of hematopoietic cells, in MM pathobiology, and in suppressing the expression of interferon-stimulating genes and T-cell stimulation. Among the CELMoDs, mezigdomide has the greatest cereblon-binding potency, plus the greatest potency for the degradation of Ikaros and Aiolos and subsequent downstream antimyeloma effects. Preclinical studies of mezigdomide have demonstrated its anti-proliferative and apoptotic effects in MM, along with its immune-stimulatory effects and its synergistic activity with other antimyeloma agents, including in lenalidomide-/pomalidomide-resistant MM cell lines and mouse xenograft models. Early-phase clinical trial data indicate notable activity in heavily pretreated patients with RRMM, including those with triple-class-refractory disease, together with a tolerable and manageable safety profile. This review summarizes current preclinical and clinical findings with mezigdomide and its potential future roles in the treatment of MM.

## 1. Introduction

Over the past 40 years, multiple myeloma (MM) has evolved from an acute disease with a median overall survival (OS) of only ~3–4 years [1] to become a chronic disease in which the median OS is now 7–10 years [2,3], and patients can require multiple lines of therapy over their disease course [4,5,6,7]. In the context of a global annual incidence of approximately 180,000 new cases [8], including more than 35,000 in the United States alone [9], and the continuing increase in survival rates [10], there is a growing population of patients requiring additional treatment options. However, given the median age at diagnosis in the United States of 69 years [10] and the potential cumulative effects of prior regimens, these new therapies need to be feasible and accessible in this patient population and require a tolerable safety profile to enable long-term treatment. Furthermore, as there is substantial heterogeneity at MM diagnosis and through the disease course associated with multiple disease-related and patient-related characteristics [4,11,12], novel therapies need to be efficacious across patient populations, including in those with high-risk features. Finally, with standards of care evolving to include quadruplet induction regimens and triplets as second-line therapies [7,13,14], there is an ongoing need for new treatment options that are active in patients who have relapsed following treatment with multiple standard classes of drugs or who may be multi-drug refractory [15].

### Current Treatment of Relapsed/Refractory MM (RRMM)

The current standards of care for the treatment of RRMM include triplet regimens comprising a monoclonal antibody (mAb) such as the anti-CD38 agents daratumumab and isatuximab and the anti-SLAM family member 7 (SLAMF7) mAb elotuzumab, plus a proteasome inhibitor (PI; bortezomib, carfilzomib, ixazomib) or an immunomodulatory drug (IMiD; lenalidomide, pomalidomide, thalidomide) and dexamethasone. For example, triplet standards of care as second-line therapy include daratumumab or isatuximab in combination with carfilzomib or pomalidomide and dexamethasone, as well as elotuzumab plus lenalidomide or pomalidomide and dexamethasone [7,13,14,16]. Regimens incorporating novel targeted agents are also recommended as RRMM treatment, including the exportin-1 (XPO1) inhibitor selinexor plus bortezomib and dexamethasone (Vd) [17] and, in Europe, the cytotoxic drug–peptide conjugate melflufen plus dexamethasone [18]. More recently, novel immune-based therapies have been approved and are being investigated early in the RRMM treatment algorithm [4,16,19,20,21], including the chimeric antigen receptor (CAR) T-cell therapies idecabtagene vicleucel (ide-cel) [22,23] and ciltacabtagene autoleucel (cilta-cel) [24,25]; the bispecific antibodies/T-cell engagers teclistamab [26], talquetamab [27], and elranatamab [28]; and, in Europe, the antibody–drug conjugate belantamab mafodotin [29].

However, the real-world applicability of therapies is a key consideration when selecting RRMM treatment options [11,30,31]. It is important to acknowledge that RRMM patients are diverse, with a wide range of ages and performance statuses, as well as differing preferences and needs. Thus, while clinical efficacy and improvements in outcomes have been demonstrated in clinical trials, patient heterogeneity and comorbidities can impact outcomes in real-world settings, along with quality of life, tolerability, treatment burden, cost, and the ability to tailor treatment approaches for individual patients. For example, for some patients, the convenience and feasibility of oral therapies may be associated with longer duration of treatment and the potential for improved outcomes [11], although costs associated with the long-term administration of some of these therapies may be challenging. For other patients, despite their efficacy and limited treatment burden in some cases, the feasibility of the novel technologies that are revolutionizing the treatment of RRMM may be reduced due to distinct practical challenges and limitations of access, such as the need for hospitalization, specialized hospital staff, and management and supportive care for the risks of novel toxicities and infectious complications, all with substantial attendant costs. Furthermore, waiting lists may present a barrier to success—a recent multi-center analysis in the United States indicated that the estimated waiting time for CAR T-cell therapy was ~6 months [32].

In this context, there remains a need for novel, applicable, and accessible treatment options in RRMM, including agents/regimens that are active in triple-class-refractory and penta-drug-refractory/exposed disease [16]. Among the emerging therapies in this setting are the new, investigational, orally administered cereblon E3 ligase modulators (CELMoDs) iberdomide and mezigdomide, both of which have orphan drug designation status (codes 665718 and 685019) with the United States Food and Drug Administration for the treatment of MM but are not yet approved in this setting. These agents have greater cereblon-binding affinity than the IMiDs; indeed, mezigdomide (formerly CC-92480), the most potent of the CELMoDs, has recently demonstrated notable clinical activity in combination with dexamethasone in triple-class-refractory RRMM [33]. Here, we review the mechanism of action, preclinical findings, and clinical data to date for mezigdomide in RRMM.

## 2. Targeting the UPS in the Treatment of MM

Over the past 30 years, the ubiquitin–proteasome system (UPS) has emerged as a key target for MM therapies [34,35,36]; the UPS, which, in healthy cells, maintains the balance between intracellular protein synthesis and degradation, is dysregulated in MM, driving cell growth and survival through increased proteasomal degradation of critical pro-apoptotic and tumor suppressor proteins. Within this multi-enzymatic system, substrate proteins are tagged with the addition of polyubiquitin chains that are recognized by the 26S proteasome for degradation (Figure 1). Ubiquitin is first recruited by a ubiquitin-activating enzyme (E1) and attached to a ubiquitin-conjugating enzyme (E2) before forming a complex with an E3 ligase. Within the UPS, there are multiple E3 ligases—these act as receptors for different intracellular protein substrates, which are then polyubiquitinated and degraded via the proteasome [34,35,36].

The power of inhibiting the dysregulated UPS in MM was first demonstrated with the first-in-class PI bortezomib [35,36,37], which remains a backbone of MM treatment due to its substantial antimyeloma activity. As the class name suggests, bortezomib inhibits the activity of the 20S proteasome within the 26S complex, thereby preventing the degradation of all proteins processed by the proteasome, including important MM-regulatory proteins. However, bortezomib treatment may be associated with toxicities arising from proteasomal inhibition and off-target effects, and subsequently, additional PIs and other ways of achieving more disease-specific inhibition of UPS functions have been investigated and developed [34,35,36].

One such approach has been to target specific E3 ligases and substrate proteins. Several technologies are in development for manipulating the UPS to achieve targeted protein degradation by bringing a target protein of interest into close proximity with an E3 ubiquitin ligase, some of which have now entered clinical trials [38]. As noted, multiple E3 ligases are involved in protein degradation by the UPS, including the cullin-RING ligases (CRLs), which incorporate a variety of substrate recruiters or receptors in order to degrade a range of substrate proteins (Figure 1). One of these receptors is cereblon, which is part of the cullin 4A–cereblon CRL (CRL4CRBN) E3 ubiquitin ligase complex that plays a key role in the ubiquitination of select target proteins (Figure 1) [38], and modulating cereblon has emerged as an extremely valuable mechanism of antimyeloma activity.

### CELMoDs

It has been demonstrated that the IMiDs, namely thalidomide [39], lenalidomide [40], and pomalidomide [41], require cereblon for their antimyeloma activity [42]. The next-generation agents that modulate cereblon are known as the CELMoDs, and these include iberdomide and mezigdomide [43]. All the IMiD and CELMoD compounds modulate/co-opt cereblon within the CRL4CRBN E3 ubiquitin ligase complex to promote increased degradation of target proteins by the UPS.

As shown in Figure 1, using the example of mezigdomide, the CELMoDs act by altering the conformation of cereblon within the E3 ligase substrate receptor, switching it from its normal ‘open’ conformation to a ‘closed’ conformation [44]. This conformational change results in the CRL4CRBN E3 ligase recruiting novel protein substrates for selective degradation by the proteasome [45], including two critical lymphoid transcription factors, Ikaros family zinc finger proteins 1 and 3 (IKZF1 and IKZF3), also known as Ikaros and Aiolos [45,46]. Ikaros and Aiolos both play key roles in the development and differentiation of hematopoietic cells, as well as in MM pathobiology [46,47], and also have important functions in suppressing the expression of interferon-stimulating genes (ISGs), including *CD38* [48], and in suppressing T-cell stimulation [49].

Ikaros and Aiolos are only recruited as substrates by the closed conformation of cereblon [44], and the percentage of cereblon in the closed conformation thus distinguishes the CELMoDs in terms of degradation efficiency and antimyeloma activity. The cereblon-binding potency of the CELMoDs iberdomide and mezigdomide is enhanced compared to the IMiD compounds associated with differences in structural interaction, resulting in increased interaction with cereblon outside of the thalidomide-binding pocket, as well as due to their administration as a single S isomer that binds more efficiently than the mixture of S and R isomers that exist with the IMiD compounds [43,50]. Of the two, the cereblon-binding affinity of mezigdomide is greater than that of iberdomide, with IC_50_ values of ~0.03 and ~0.06 μM, respectively. Furthermore, differences in how the compounds bind translate into a greater percentage of cereblon in the closed (or ‘active’) confirmation with the CELMoDs—while only 20% of cereblon exists in the closed confirmation with pomalidomide at saturating concentrations, this increases to 50% with iberdomide and 100% with mezigdomide. Consequently, mezigdomide has the greatest potency for the degradation of Ikaros and Aiolos and the subsequent downstream antimyeloma effects [43,51]. Of note, this mechanism has been shown to drive the activity of CELMoDs, and mezigdomide in particular, in other indications including in models of acute myeloid leukemia [52] and lenalidomide-resistant T-cell lymphomas [53,54].

For a more comprehensive review of the mechanism of action of CELMoDs, see the paper by Barankiewicz et al. in this journal [55].

## 3. Preclinical Rationale for Activity of Mezigdomide in RRMM

The immune effects of the IMiDs are well established and include increased interleukin (IL)-2 and interferon-γ production, T-cell proliferation, and natural killer (NK) and NK T-cell activation, counteracting the immune dysfunction effects caused by the development of MM and resulting in substantial antimyeloma activity, notably in combination with anti-CD38 and anti-SLAMF7 mAbs [56]. With the enhanced potency of the latest CELMoDs, iberdomide and mezigdomide, and the resultant enhanced protein degradation compared to the older IMiDs, both agents have demonstrated greater antitumor activity through direct apoptotic effects and greater immunomodulatory activity against MM cells [50,57], including increased IL-2 secretion and cytotoxic T-cell infiltration [58,59]. Furthermore, iberdomide has been shown to disrupt the tumor microenvironment in patients with RRMM, resulting in significant increases in effector T and NK cells [59]—these mechanisms of adaptive and innate immune enhancement in the bone marrow have also been reported recently with mezigdomide, with immune cell populations shifting from exhausted/senescent to activated following mezigdomide treatment [60], associated with Ikaros/Aiolos degradation [61]. Overall, both iberdomide and mezigdomide offer improved profiles compared to lenalidomide and pomalidomide in terms of cereblon binding, specificity as a single S isomer, targeted protein degradation, tumor anti-proliferation, tumor apoptosis, immune stimulation, and synergistic combinability (Figure 2) [43,50,57,62], with these aspects being suggestive specifically of a unique clinical profile for mezigdomide in MM. Importantly, mezigdomide has been shown to have apoptotic effects in lenalidomide-resistant/refractory and pomalidomide-resistant/refractory cell lines [57], providing the rationale for its evaluation in this setting in the clinic.

A number of preclinical evaluations have demonstrated enhanced antimyeloma and immunomodulatory effects with mezigdomide in combination with a range of agents used for the treatment of MM. Initial studies demonstrated synergistic in vitro cell killing and significantly greater tumor growth inhibition in vivo in a lenalidomide-resistant xenograft mouse model with mezigdomide combined with dexamethasone or bortezomib compared with individual agents alone [63]. Additional preclinical studies of mezigdomide in combination with PIs have demonstrated potent immunomodulation with mezigdomide plus bortezomib and no adverse impact on the immunostimulatory effects of mezigdomide in combination with bortezomib [64]. Furthermore, in combination with bortezomib or carfilzomib, mezigdomide resulted in significantly greater in vitro apoptotic activity than pomalidomide [65,66] at a 100-fold lower concentration, while in vivo data showed near-complete tumor regressions with mezigdomide plus Vd and prolonged survival in mouse models compared to pomalidomide plus Vd [66]. These findings provide the rationale for the investigation of mezigdomide–PI–dexamethasone triplet regimens in the clinic.

It has also been shown that mezigdomide increases CD38 cell surface expression in MM cell lines, leading to enhanced antibody-dependent cellular cytotoxicity (ADCC) and antibody-dependent cellular phagocytosis (ADCP) with subsequent daratumumab exposure [63]. This ‘priming’ effect and consequent synergy with anti-CD38 mAbs may arise as a result of mezigdomide potently mediating the degradation of Ikaros and Aiolos, which suppress CD38 expression [48]. Additionally, studies have shown synergistic antitumor activity with mezigdomide and daratumumab and significantly greater apoptosis than with either agent alone [67].

Similar sensitization and synergistic effects have been reported in preclinical studies of mezigdomide in combination with bispecific antibodies. For example, mezigdomide pretreatment was shown to potentiate the cytotoxicity of the B-cell maturation antigen (BCMA)xCD3 bispecific antibody alnuctamab [68], which is among those being evaluated in patients with RRMM [69]. Mezigdomide upregulated the expression of adhesion molecules in MM cells and peripheral blood mononuclear cells (PBMCs) that are key for T-cell-mediated cell killing, thereby enhancing T-cell/target interactions and the antimyeloma effects of alnuctamab [68]. In co-culture experiments with a range of MM cell lines, mezigdomide pretreatment resulted in greater enhancement of alnuctamab activity than iberdomide or pomalidomide [58]; in vivo, in a humanized mouse MM xenograft model, the antitumor activity of alnuctamab was also enhanced by mezigdomide pretreatment or concurrent administration, with increased T-cell activation and infiltration of tumor tissue. Furthermore, mezigdomide in combination with the G protein-coupled receptor 5D (GPRC5D)xCD3 bispecific antibody forimtamig resulted in rapid tumor regressions and significantly better progression-free survival (PFS) rates than with forimtamig alone in vivo [70].

An interesting observation in the context of bispecific antibody therapy for RRMM is that CELMoDs can reduce proinflammatory cytokine secretion, including IL-6 and IL-1β, from monocytes/macrophages and PBMCs in vitro [71]. The greatest effects were seen with mezigdomide pretreatment, which suppressed the secretion of IL-6 and IL-1β induced by alnuctamab exposure. These findings suggest a potential role for mezigdomide in mitigating cytokine release syndrome (CRS), which is a common toxicity associated with bispecific antibodies [71].

Preclinical work has also elucidated a number of other mechanism-related effects associated with mezigdomide that may have clinical relevance. As a downstream effect of promoting Ikaros and Aiolos degradation, CELMoDs have been shown to downregulate the expression of CDC28 protein kinase regulatory subunit 1B (CKS1B), which is associated with poor prognosis in MM through its role in chromosome 1q amplification [72]. Through this mechanism, lenalidomide, pomalidomide, and mezigdomide in combination with a bromodomain-containing protein 4 (BRD4) inhibitor resulted in synergistic decreases in MM cell proliferation and increases in apoptosis [72], suggesting the potential for mezigdomide to have specific activity in MM with gain/amplification of 1q21 via mediation of proteasomal degradation of CKS1B. Studies have also established a number of potential mechanisms of resistance to CELMoDs [73] that result in reduced cereblon protein expression and thus reduced activity [74]; for example, *CRBN* alterations or copy number loss driven by mutations or messenger RNA (mRNA) splice variants lacking exon 10, together with monoallelic 3p26 loss, have been suggested to mediate resistance to mezigdomide and IMiDs, as well as decreased expression of the UPS-regulating constitutive photomorphogenesis 9 (COP9) signalosome protein complex [75,76,77]. Furthermore, overexpression of ubiquitin-specific peptidase 15 (USP15), which antagonizes the ubiquitination of substrate proteins on CRL4CRBN E3 ligase, is also associated with IMiD resistance [78]. Potential biomarkers such as these may prove valuable in understanding CELMoD sensitivity or resistance in patients with RRMM; however, these results are based on a very limited number of patients and will require confirmation in larger analyses.

## 4. Clinical Data from Studies of Mezigdomide in RRMM

At the time of writing, data have been published or reported from the first two clinical trials of mezigdomide in RRMM, alone or in combination with dexamethasone and with Vd, carfilzomib–dexamethasone (Kd), daratumumab–dexamethasone (Dd), and elotuzumab–dexamethasone (Ed) [33,79,80]. Additionally, phase 3 studies are underway.

### 4.1. Clinical Safety Profile of Mezigdomide

The safety and tolerability of mezigdomide in patients with RRMM were first explored in the dose-escalation phase 1 component of the first-in-human CC-92480-MM-001 phase 1/2 study of mezigdomide plus dexamethasone [33]. A total of 77 patients were treated within 13 dose cohorts (0.1–2.0 mg once daily or 0.2–0.8 mg twice daily) across four dosing schedules that included dosing for 3, 7, or 10 of every 14 days, or for 21 of every 28 days. Of these, 24 patients received mezigdomide 0.8 or 1.0 mg once daily for 21 of every 28 days, plus weekly dexamethasone, including 11 treated at the phase 2 dose of 1.0 mg. Overall, the patients were heavily pretreated, with a median of 6 (range 2–13) prior lines of therapy; 43 (56%) were triple-class (PI, IMiD, anti-CD38 mAb)-refractory, and 23 (30%) had high-risk cytogenetics [comprising del17p, t (4;14), t (14;16), amp1q21 (≥4 copies)].

The most common all-grade and grade 3/4 adverse events (AEs) reported in the phase 1 component of the study were neutropenia (80.5%, grade 3/4 71.5%) and infections (74.0%, grade 3/4 40.3%) (Figure 3), followed by anemia (61.0%, grade 3/4 37.7%) and thrombocytopenia (50.6%, grade 3/4 23.4%). These toxicities reflect the mechanism of action of mezigdomide; notably, neutropenia is a specific mechanism-mediated AE, arising from the role of Ikaros and Aiolos in granulocyte maturation [44]. Other than infections, the most common non-hematologic AEs were fatigue (40%) and nausea (27%), with only fatigue (10%) reported at grade 3/4 severity in ≥5% of patients. Overall, 19 (24.7%) patients required mezigdomide dose reductions due to AEs [33].

As expected, this safety profile of mezigdomide plus dexamethasone was reflected at the selected phase 2 dose [33]. A total of 101 patients were treated at this dose, all of whom had triple-class-refractory disease after a median of 6 (range 3–15) prior lines of therapy, which included anti-BCMA therapy in 30 (29.7%) patients; 37 (36.6%) had high-risk cytogenetics and 40 (39.6%) had plasmacytomas. Common AEs were consistent with phase 1 findings, with neutropenia (77.2%, grade 3/4 75.2%), infections (65.3%, grade 3/4 34.7%), anemia (52.5%, grade 3/4 35.6%), and thrombocytopenia (42.6%, grade 3/4 27.7%) again being the most frequently reported (Figure 3). Other common non-hematologic AEs were fatigue (36%) and diarrhea (31%), with only fatigue and dyspnea (each 5%) reported at grade 3/4 severity in ≥5% of patients. Similarly to the phase 1 experience, 29 (28.7%) patients required mezigdomide dose reductions due to AEs [33]. Notably, in this heavily pretreated population, the rate of mezigdomide discontinuation due to AEs was low, at 5.9% (*n* = 6), with patients receiving a median of 4 and up to 20 cycles of treatment. Neutropenia was managed through the use of granulocyte colony-stimulating factor (G-CSF), which was administered to 78 (77.2%) patients overall, including prophylactically in 47 (46.5%). A consistent safety profile was seen with mezigdomide alone at a dose of 0.4 mg (*n* = 5) or 0.6 mg (*n* = 12) in preliminary data from the CC-92480-MM-001 study, with grade 3/4 neutropenia (82.4%), anemia (41.2%), thrombocytopenia, and infections (each 17.6%) being among the most common toxicities, with low rates of non-hematologic grade 3/4 AEs, in triple-class-exposed/refractory patients with RRMM [80].

In the ongoing CC-92480-MM-002 phase 1/2 study, 104 patients with RRMM have been enrolled to date to receive mezigdomide plus Vd and mezigdomide plus Kd. These patients were less heavily pretreated than those in the study of mezigdomide plus dexamethasone, with medians of 1–3 prior therapies (overall range 1–4) across the study cohorts and with 20 (19.2%) patients being triple-class refractory, but were more likely to have high-risk cytogenetics (*n* = 55, 52.9%) [79]. The preliminary safety profiles were similar to those seen with mezigdomide plus dexamethasone, with neutropenia and infections consistently shown to be the most common all-grade and grade 3/4 AEs (Figure 3), followed by anemia and thrombocytopenia, and relatively low rates of other grade 3/4 non-hematologic AEs [79]. As with mezigdomide plus dexamethasone, neutropenia was manageable with dose interruptions and G-CSF, and the rate of discontinuations due to AEs was low [79]. However, it should be noted that patient numbers are somewhat limited in some of the cohorts, and two of those that have been reported to date are dose-escalation cohorts. In a further two cohorts of this study, as reported at the 2023 Annual Meeting of the American Society of Hematology (ASH), 57 patients to date have received mezigdomide plus Dd after a median of two prior regimens and 20 have received mezigdomide plus Ed after a median of three prior regimens [81]. The preliminary safety profiles of these regimens were consistent with the others, with the most common grade 3/4 events being neutropenia (53.6% and 40.0%) and infections (19.6% and 35.0%) with mezigdomide plus Dd or Ed, respectively [81].

### 4.2. Clinical Pharmacokinetics and Pharmacodynamics of Mezigdomide

Data from the phase 1/2 study of mezigdomide plus dexamethasone have shown that mezigdomide has dose-dependent pharmacokinetics, with increasing systemic exposure following consecutive dosing for at least 5 days [33]. A linear dose–exposure relationship was similarly seen in a population pharmacokinetic model for mezigdomide developed based on data collected in healthy subjects [82]; this model will be informative for dosing guidelines for mezigdomide, as it also demonstrated a ~30% increase in oral bioavailability when mezigdomide was taken following a high-fat meal and a ~64% reduction when taken in conjunction with proton-pump inhibitors—thus, co-administration of mezigdomide with proton-pump inhibitors is not recommended [82].

The pharmacodynamic effects of mezigdomide have also been demonstrated in the phase 1/2 study of mezigdomide plus dexamethasone; mezigdomide dosing resulted in maximal Aiolos degradation after 3 h, with effects sustained through 24 h, supporting the selection of once-daily rather than twice-daily dosing [33]. This pharmacodynamic effect was also seen in tumor samples (bone marrow plasma cells) from patients with RRMM, including those who had received pomalidomide in their last prior regimen [33,83]. Substantial and near-complete reductions from baseline were demonstrated. Aiolos levels were then seen to recover during dosing schedule windows, supporting the use of a 21 out of 28-day schedule for sustained Aiolos suppression [33]. Similarly, substantial reductions from baseline in Aiolos levels have been seen with mezigdomide plus Vd or Kd in preliminary data from the ongoing phase 1/2 study of these triplet regimens [7].

Pharmacodynamic analyses have also shown the effects on the immune system of mezigdomide dosing. Post-baseline expansion of proliferating, activated CD4+, activated CD8+, and effector memory T cells was demonstrated in samples collected in the phase 1/2 study of mezigdomide plus dexamethasone, indicating a shift from a naïve to an effector phenotype, the dynamics of which were aligned with Ikaros and Aiolos degradation due to mezigdomide treatment [33,83]. Post-baseline increases in proliferating T cells have also been seen with mezigdomide plus Vd or Kd [7], and immune stimulation of T and NK cells has been reported with mezigdomide plus Dd [67,81]. Such immune-stimulating effects may be valuable in developing rational combination regimens, including with immune effector cell therapies.

### 4.3. Clinical Efficacy of Mezigdomide

The response rates achieved with mezigdomide plus dexamethasone in the phase 1/2 clinical study [33] were notable given the heavily pretreated patient population. An overall response rate (ORR) of 24.7% was seen in the phase 1 dose-escalation component of the study, which included responses in 6 of 11 (54.5%) patients treated with mezigdomide 1.0 mg on a 21-day schedule, the initial dose and schedule used in phase 2. Among the 101 triple-class-refractory patients treated in the phase 2 component, the ORR was 40.6%, which included 24.8% with a very good partial response (VGPR) or better (Figure 4). Even in poor-prognosis patients with plasmacytomas or high-risk cytogenetics, ORRs of 30.0% and 32.4%, respectively, were achieved with mezigdomide plus dexamethasone, while 50% of patients previously treated with anti-BCMA therapy responded to the doublet regimen, indicating substantial activity after multiple previous classes of therapy [33]. Additionally, preliminary data from the CC-92480-MM-001 study have shown an ORR of 50.0% with mezigdomide alone in 12 patients treated at a dose of 0.6 mg, which included two VGPRs and four partial responses; no responses were seen in 5 patients treated at 0.4 mg [80].

Importantly, although the overall median PFS in the phase 2 population was 4.4 months, durable benefit was seen in those patients who responded to mezigdomide plus dexamethasone. The median duration of response (DOR) was 7.6 months in the 41 responders, of whom 10 were ongoing on treatment at the data cut-off for the publication and 5 had remained in response for ≥12 months [33]; the median DOR in patients who received the phase 2 dose in the phase 1 component was 9.2 months. Durable responses were seen in poor-prognosis patients, with a median DOR of 10.0 months in responding patients with high-risk cytogenetics, while in patients who had previously received anti-BCMA therapy, the median DOR was 6.9 months [33]. These data demonstrate that notable clinical benefit was seen in a subset of triple-class-refractory patients in the phase 2 component, particularly in the context of expected outcomes in this population [84,85], with a median DOR longer than that seen in this setting with selinexor–dexamethasone [86] and with melflufen–dexamethasone [87].

More substantial efficacy has been seen in preliminary data from the second clinical study investigating mezigdomide in combination with Vd, Kd, Dd, or Ed in less heavily pretreated patients with RRMM [79,81]. As shown in Figure 4, ORRs of 75.0–90.9% have been reported to date with mezigdomide plus Vd or Kd triplet regimens, including ≥VGPR rates of 39.3–81.8% and rates of complete response (CR) or stringent CR of 14.8–27.3% [79]. As with mezigdomide plus dexamethasone, these responses appear durable in a subset of patients. Median DOR was 10.9 months and not reached in the dose-escalation and expansion cohorts receiving mezigdomide plus Vd, respectively, and 12.3 months in the dose-escalation cohort receiving mezigdomide plus Kd, with some patients receiving treatment for up to nearly 4 years and 27 (26.0%) ongoing on treatment at data cut-off [79]. Notably, both triplet regimens appeared active in patients refractory to prior IMiD therapies, with ORRs of 76.9% and 83.3% with mezigdomide plus Vd and Kd, respectively, in 13 and 12 patients refractory to prior pomalidomide, and ORRs of 69.2% (mezigdomide plus Vd dose-escalation cohort), 75.0% (mezigdomide plus Vd expansion cohort), and 82.4% (mezigdomide plus Kd dose-escalation cohort) in 13, 12, and 17 patients, respectively, refractory to prior lenalidomide and anti-CD38 mAbs [79]. Such activity is of importance in the context of quadruplet regimens incorporating both lenalidomide and an anti-CD38 mAb emerging as standards of care in the frontline setting [7]. Preliminary data on mezigdomide plus Dd or Ed reported at ASH 2023 indicated ORRs of 75.0% and 45.0%, respectively, with a ≥VGPR rate of 46.4% with mezigdomide plus Dd (Figure 4), indicating promising efficacy in this dose-evaluation stage [81].

## 5. Discussion and Future Perspectives

Current clinical data are demonstrating substantial activity with mezigdomide-based regimens in the treatment settings investigated to date, together with a manageable safety profile [33,79]. Development of mezigdomide therapy for RRMM is mirroring that for pomalidomide, with initial dexamethasone-based doublet studies in patients refractory to prior IMiD therapy [88,89,90] being followed by evaluation of PI–dexamethasone-based triplets [91,92,93] and then mAb–dexamethasone-based triplets [94,95,96]. With regard to the latter combinations, preliminary data on mezigdomide plus Dd or Ed have been reported from the ongoing CC-92480-MM-002 phase 1/2 study (NCT03989414) [81]. This is a platform study covering multiple regimens, with additional cohorts planned to evaluate different schedules of mezigdomide plus Dd, as well as mezigdomide plus isatuximab and dexamethasone (Figure 5A). Overall, an estimated 424 patients will be enrolled across the various cohorts of this study, the mature data from which will help determine whether the synergistic effects of mezigdomide and mAbs in preclinical investigations [63] are borne out in the clinic.

Additionally, there are two large phase 3 randomized studies ongoing in patients with RRMM (Figure 5B). Building on existing data with mezigdomide plus Vd, the SUCCESSOR-1 trial is evaluating mezigdomide versus pomalidomide in combination with Vd in lenalidomide-exposed patients with one to three prior therapies; an initial study stage will evaluate three different doses of mezigdomide and the optimal dose will then be used in the second stage in a head-to-head comparison with pomalidomide plus Vd [97]. Similarly, the SUCCESSOR-2 trial will build on initial data with mezigdomide plus Kd to evaluate this triplet regimen against the standard Kd doublet in patients with one or more prior therapies who have received prior lenalidomide and an anti-CD38 mAb—again, an initial dose-optimization stage will evaluate mezigdomide at three different doses prior to selection of the optimal dose to be used in the second stage of the study [98].

These and other studies will help determine the future role(s) of mezigdomide-based therapy in the treatment of RRMM, for which the unmet needs and areas of opportunity continue to evolve with changes in frontline standards of care and the introduction of new treatment options such as immune cell effector therapies. Data to date suggest that mezigdomide regimens may play an important role in treating patients who are refractory to prior IMiDs, PIs, and mAbs, and, due to its potency and mechanism of action, mezigdomide may emerge as a valuable combination partner for existing and novel therapies, including immune effector cell therapies such as bispecific antibodies and CAR T-cell therapies.

## 6. Conclusions

As the most potent CELMoD, mezigdomide results in the rapid degradation of Ikaros and Aiolos substrate proteins and induction of apoptosis in MM cells, together with strong stimulation of the immune system, and thus has a differentiated preclinical profile compared to the IMiD compounds. In the context of preclinical and clinical activity in the setting of IMiD-refractory MM, mezigdomide has the potential to treat advanced disease with continued immunomodulatory activity, including in poor-prognosis settings such as triple-class-refractory disease, high-risk cytogenetics, the presence of extramedullary disease, and prior exposure to anti-BCMA therapy. Clinical findings to date show notable activity in combination with dexamethasone alone and with Vd, Kd, Dd, or Ed, with a manageable safety profile and the convenience of an orally administered therapy, and ongoing investigations will further determine its feasibility and activity as a combination partner for mAbs in RRMM. While the IMiD compounds lenalidomide and pomalidomide remain among the backbones of care for newly diagnosed MM and RRMM, mezigdomide and the other novel CELMoD iberdomide are being extensively studied and may emerge in due course as additional treatment options to help contribute towards improved overall patient outcome.

## Figures and Tables

**Figure 1 cancers-16-01166-f001:**
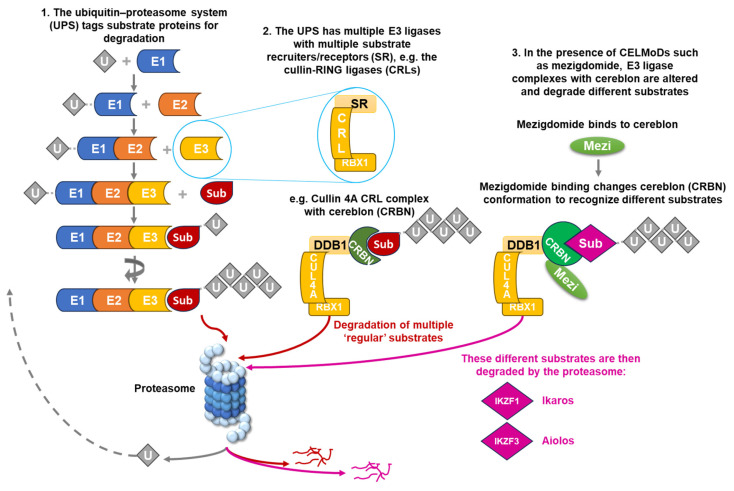
The ubiquitin–proteasome system, the cullin 4A–cereblon E3 ligase complex, and the effects of CELMoDs such as mezigdomide. CELMoD, cereblon E3 ligase modulator; CRBN, cereblon; CRL, cullin-RING ligase; CUL4A, cullin 4A; DDB1, DNA damage-binding protein 1; Mezi, mezigdomide; RBX1, RING box protein-1; SR, substrate recruiter/receptor; Sub, substrate protein; U, ubiquitin; UPS, ubiquitin–proteasome system.

**Figure 2 cancers-16-01166-f002:**
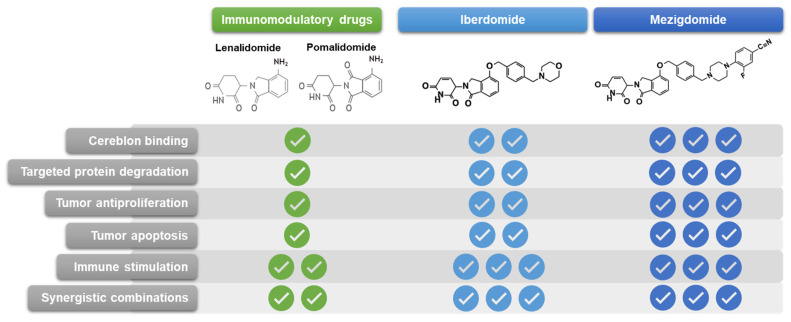
Chemical structures and relative preclinical properties of immunomodulatory drugs, iberdomide, and mezigdomide.

**Figure 3 cancers-16-01166-f003:**
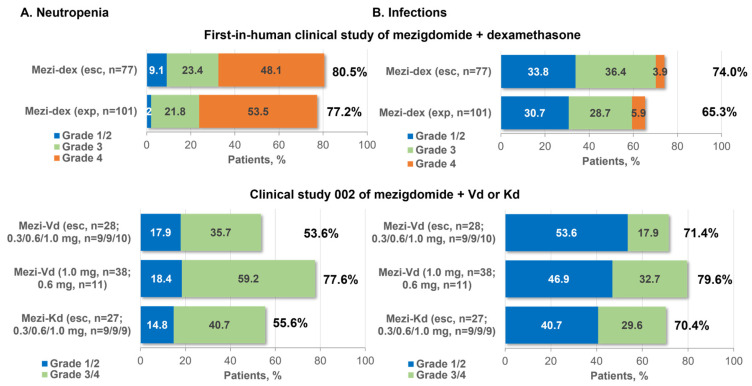
Incidence of neutropenia and infections with mezigdomide-based combination therapy in the first two clinical trials in RRMM. Esc, dose-escalation cohort; exp, expansion cohort; Kd, carfilzomib–dexamethasone; Mezi, mezigdomide; RP2D, recommended phase 2 dose; Vd, bortezomib–dexamethasone.

**Figure 4 cancers-16-01166-f004:**
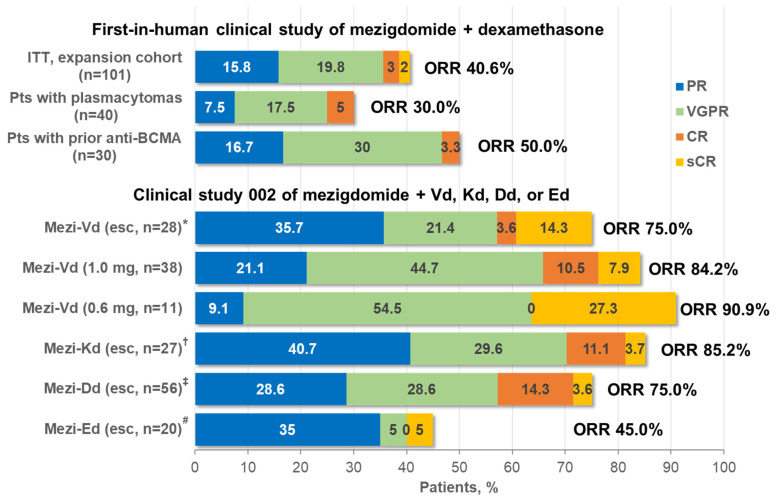
Clinical responses with mezigdomide-based combination therapy in the first two clinical trials in RRMM. * Mezigdomide doses of 0.3, 0.6, and 1.0 mg in 9, 9, and 10 patients, respectively. ^†^ Mezigdomide doses of 0.3, 0.6, and 1.0 mg in 9, 9, and 9 patients, respectively. ^‡^ Mezigdomide doses of 0.3 and 0.6 mg in 20 and 36 patients, respectively. ^#^ Mezigdomide doses of 0.3 and 0.6 mg in 11 and 9 patients, respectively. BCMA, B-cell maturation antigen; CR, complete response; Dd, daratumumab–dexamethasone; Ed, elotuzumab–dexamethasone; esc, dose-escalation cohort; ITT, intent-to-treat; Kd, carfilzomib–dexamethasone; Mezi, mezigdomide; ORR, overall response rate; PR, partial response; Pts, patients; RP2D, recommended phase 2 dose; sCR, stringent complete response; Vd, bortezomib–dexamethasone; VGPR, very good partial response.

**Figure 5 cancers-16-01166-f005:**
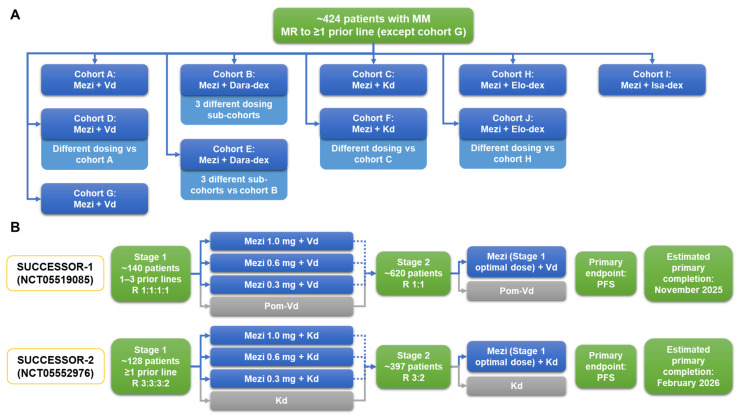
Ongoing (**A**) phase 1/2 platform study of multiple mezigdomide-based combination regimens and (**B**) phase 3 SUCCESSOR-1 [97] and SUCCESSOR-2 [98] trials of mezigdomide in RRMM. Additionally, the combination of Mezi + Elo-dex is being evaluated in a phase 1 study in patients with RRMM who have received CD38- and BCMA-targeted therapies (NCT05981209), for which the estimated primary completion date is December 2024 (ClinicalTrials.gov, accessed on 18 September 2023). BCMA, B-cell maturation antigen; Dara, daratumumab; dex, dexamethasone; Elo, elotuzumab; Isa, isatuximab; Kd, carfilzomib–dexamethasone; Mezi, mezigdomide; MM, multiple myeloma; MR, minimal response; PFS, progression-free survival; RRMM, relapsed/refractory multiple myeloma; Vd, bortezomib–dexamethasone.

## Data Availability

The data presented in this article are available in the cited sources.

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
