# Peer review of "Mezigdomide—A Novel Cereblon E3 Ligase Modulator under Investigation in Relapsed/Refractory Multiple Myeloma"

_cancers, 2024, doi:10.3390/cancers16061166_

Round 1

Reviewer 1 Report

Comments and Suggestions for Authors

A minor concern is references that some of them are not originally described, such as CKS1B. 

Author Response

We thank the reviewer for highlighting this issue; we have checked abbreviations throughout the manuscript and defined at first use those that are not already defined, including CKS1B.

Reviewer 2 Report

Comments and Suggestions for Authors

This review is timely interesting and well organised. However, it doesn't have any novelty since this grroup published similar paper entitled, " The role of E3 ubiquitin ligase in multiple myeloma: potential for cereblon E3 ligase modulators in the treatment of relapsed/refractory disease.

Richardson PG, Mateos MV, Vangsted AJ, Ramasamy K, Abildgaard N, Ho PJ, Quach H, Bahlis NJ.Expert Rev Proteomics. 2022 Apr-Jun;19(4-6):235-246. doi: 10.1080/14789450.2022.2142564. Epub 2022 Nov 9." and 
Mezigdomide and Multiple Myeloma published in NEJM in 2023.

Author Response

We acknowledge the reviewer’s point regarding the previous review paper and the recent primary study publication. However, our review provides comprehensively updated information on preclinical and clinical studies of mezigdomide that was not available at the time of the previous review, and it also presents an extensive summary of clinical findings beyond those reported in the primary study publication in the New England Journal of Medicine, including data from recent congress presentations. We believe our paper thus provides novelty as a ‘bench-to-bedside’ review of the most potent investigational agent in the CELMoD class.

Reviewer 3 Report

Comments and Suggestions for Authors

The authors report mezigdomide- a novel cereblon E3 ligase modulator for the treatment of relapsed/refractory multiple myeloma. However, mezigdomide has not approved yet, thus, it is not appropriate to report as a review.

Author Response

We acknowledge the reviewer’s concern that mezigdomide has not been approved yet by either the United States Food and Drug Administration or the European Medicines Agency. We have therefore highlighted this fact more clearly by stating that it is an investigational agent and noting the orphan drug designation status for the treatment of multiple myeloma of both iberdomide and mezigdomide. We believe that a summary of the potential for an agent in this area of unmet clinical need is important, and note that reviews of other investigational agents for relapsed/refractory multiple myeloma have been published in the absence of regulatory approval, on e.g. venetoclax (PMID: 38149670, 38113108), CAR T cell therapies ide-cel and cilta-cel (PMID: 32978608, 32850376), pomalidomide (PMID: 20095057), melflufen (PMID: 29029544, 32924646, 32992506), isatuximab (PMID: 31779273), and panobinostat (PMID: 22404247).

Reviewer 4 Report

Comments and Suggestions for Authors

This review presented evidence from pre-clinical studies and clinical trials to prove that mezigdomide is a promising CELMoD to treat RRMM. It included all important literatures about mezigdomide and the writing is quite good. I have nothing to suggest to make it better.

Author Response

We thank the reviewer for their kind comments.

Round 2

Reviewer 2 Report

Comments and Suggestions for Authors

Much improved

Reviewer 3 Report

Comments and Suggestions for Authors

none